

# Population structure and fecundity of the Xanthid crab *Leptodius exaratus* (H. Milne Edwards, 1834) on the rocky shore of Gujarat state, India

Krupal Patel[1,*], Heris Patel[1,*], Swapnil Gosavi[2], Kauresh Vachhrajani[2] and Jigneshkumar Trivedi[1]

[1] Department of Life Sciences, Hemchandracharya North Gujarat University, Patan, Gujarat, India
[2] Department of Zoology, The Maharaja Sayajirao University of Baroda, Vadodara, Gujarat, India
* These authors contributed equally to this work.

## ABSTRACT

**Background:** The population structure and breeding biology of the Xanthid crab, *Leptodius exaratus* (H. Milne Edwards, 1834), on the rocky intertidal region of Shivrajpur in Saurashtra coast, Gujarat state, were examined.

**Method:** From March 2021 to February 2022, monthly sampling was conducted during low tide using catch per unit effort in the 500 m$^2$ area. The sampled specimens were categorised into male, non-ovigerous female or ovigerous female. In order to estimate fecundity, the morphology of the crab specimens (carapace width and body weight) as well as the size of eggs, number of eggs and weight of egg mass were recorded.

**Results:** A total of 1,215 individuals were sampled of which 558 individuals were males and 657 individuals were females. The size (carapace width) of males ranges from 5.15 to 29.98 mm, while females ranges from 5.26 to 28.63 mm which shows that the average size of male and female individuals did not differ significantly. The overall as well as monthly sex ratio was skewed towards males with a bimodal distribution while unimodal in females. The population breeds year-round, which was indicated by the occurrence of ovigerous females throughout the year. However, the maximum percentage occurrence of ovigerous females was observed from December to April which indicates the peak breeding season. The size of eggs, number of eggs and weight of egg mass were shown to positively correlate with the morphology of ovigerous females (carapace width and wet weight).

## INTRODUCTION

Investigation on the population structure of intertidal crabs started in early 1940s (*Flores & Paula, 2002*), which can reveal the patterns of species interactions and their roles within ecosystems. Accounts on population structure and breeding biology majorly try to understand the genetic diversity, age, spatial distribution, abundance, sex ratio, variation in year-round composition, fecundity of the species, as well as juvenile recruitment (*Litulo, 2005*; *Saher & Qureshi, 2010*; *Hu et al., 2015*; *Manzoor et al., 2016*). A species' life history

Corresponding author
Jigneshkumar Trivedi,
jntrivedi26@yahoo.co.in

might differ by habitat or even by location. For example, a slight variation in the latitude leading to climatic variation can cause differences between the populations. Variations in the population trends are also due to the effects of several biotic and abiotic factors affecting populations differently (*Lycett et al., 2020*). Studies on the population structure and breeding biology of a species can help determine its ecological stability in a given habitat and also contribute to our understanding of the species' biology (*Santos, Negreiros-Fransozo & Padovani, 1995*; *Takween & Qureshi, 2005*). This knowledge helps ecologists understand how different species coexist, compete, and interact, influencing ecosystem dynamics. However, such studies have not been carried out so far on some of the commonly occurring brachyuran crabs of Gujarat state.

The Saurashtra coast of Gujarat state, India, is characterised by its rocky intertidal coasts, which support a great diversity of marine organisms, including intertidal crustaceans, especially crab population. With its major inhabiting marine intertidal species, majority of the crab studies have focused on the diversity (*Trivedi & Vachhrajani, 2013a*, *2013b*, *2015*, *2016a*, *2018*; *Trivedi et al., 2015*, *2020*, *2021*; *Gosavi et al., 2017a*, *2017b*, *2021*; *Patel, Patel & Trivedi, 2020*, *2021a*; *Bhat & Trivedi, 2021*; *Padate et al., 2022*). Very less is known about the population structure of these important intertidal organisms (*Trivedi & Vachhrajani, 2016b*, *2017*; *Patel et al., 2020*; *Patel, Vachhrajani & Trivedi, 2022*). *Leptodius exaratus* (H. Milne Edwards, 1834) is a xanthid crab that is commonly found in the rocky shores of Saurashtra coast (*Patel, Patel & Trivedi, 2021b*). This crab species has been commonly reported from the rocky intertidal regions of Indo-Pacific region (*Kneib & Weeks, 1990*; *Naderloo, 2017*). It is an omnivorous species that prefers benthic fauna over algae to feed upon and is expected to have a considerable impact on how the benthic ecosystem is structured (*Al-Wazzan et al., 2020*).

In the Indian subcontinent *L. exaratus* is recorded from Andaman and Nicobar Islands, Maharashtra, Tamil Nadu, Lakshadweep Islands, Goa, Karnataka, and Gujarat (*Trivedi et al., 2018*). Though the species is very commonly found on the Saurashtra coast of Gujarat, studies only on its taxonomy (*Chopra & Das, 1937*; *Trivedi & Vachhrajani, 2015*) and colour variation (*Patel, Patel & Trivedi, 2021b*) have been carried out so far. Hence the current investigation was aimed to (1) understand the population structure and (2) study the breeding biology of *L. exaratus* occurring on the rocky shore of Saurashtra coast, Gujarat state, India in order to obtain knowledge about the ecology of rocky intertidal habitats. Studying the population structure and breeding biology of *L. exaratus* which is commonly found on the Saurashtra coast, would provide a baseline data that plays a pivotal role in understanding the effects of changing environment, habitat, or anthropogenic pressure. The present study will help in elucidating the coastal health of study area.

## MATERIALS AND METHODS

### Study area

The investigation was conducted on the rocky shore of Shivrajpur (22°19′55″N 68°57′03″E) which is located on the Saurashtra coast of Gujarat state, India (Fig. 1).
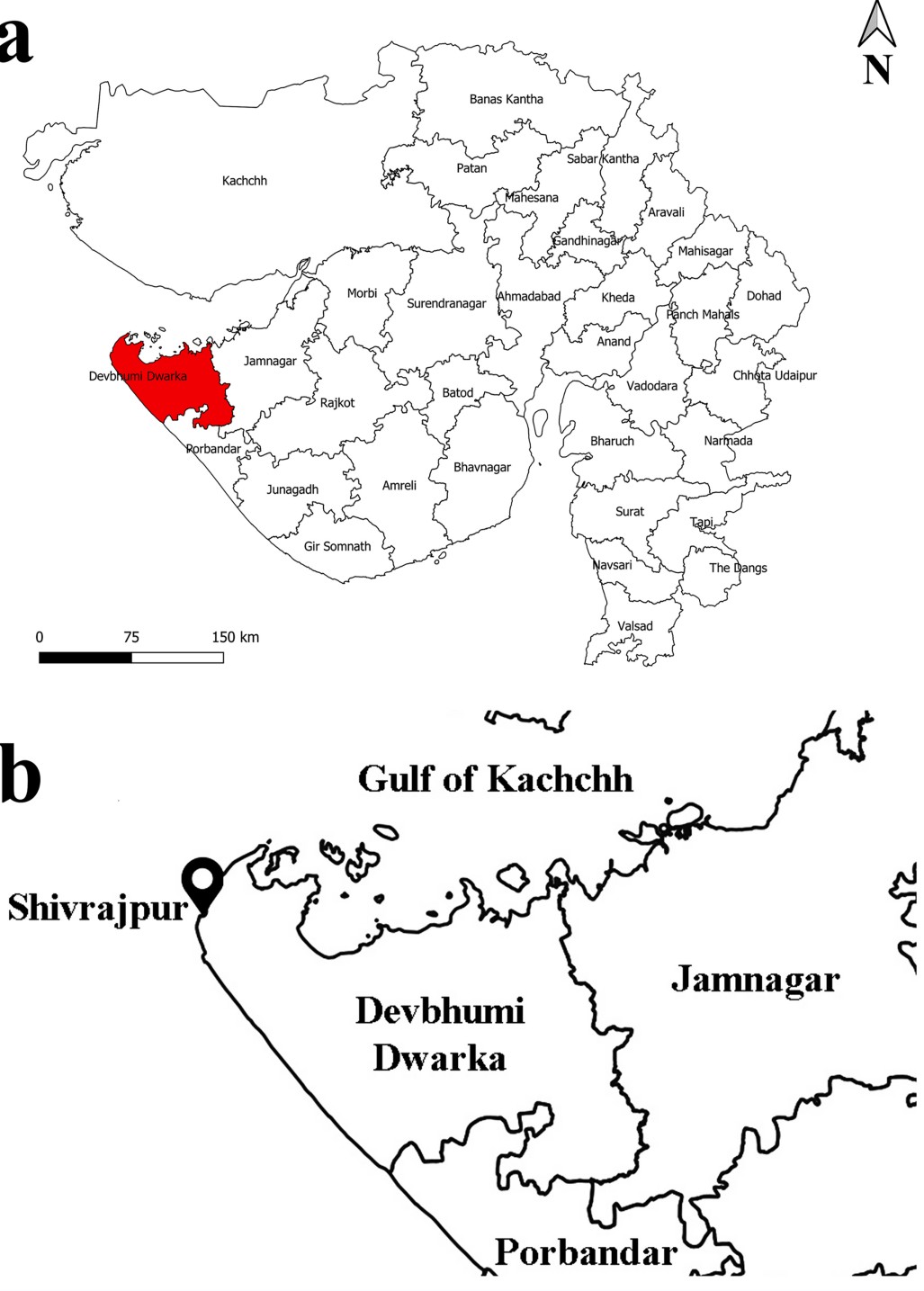

**Figure 1 Map of study area: (A) Gujarat state; (B) Shivrajpur village, Gujarat state, India.** (Prepared using QGIS version 3.14).

*Leptodius exaratus* is a small crab found abundantly in the rocky shore of the study area (Fig. 2). During low tide, the exposed area of rocky intertidal region varies from 60 to 150 m.

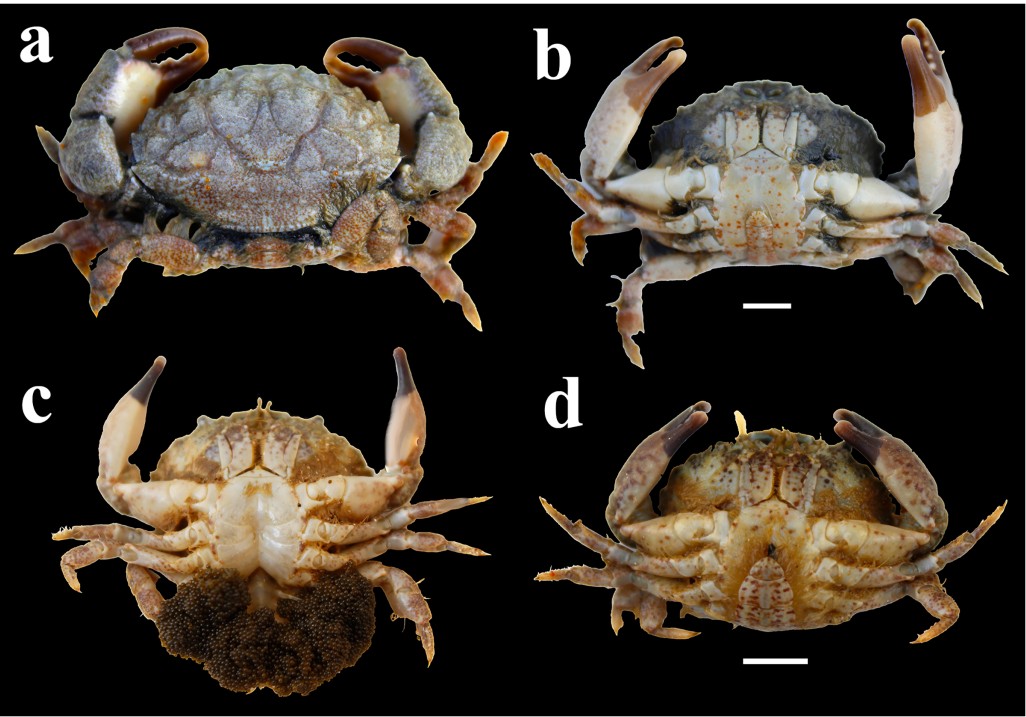

**Figure 2 Morphology of _L. exaratus_ from Shivrajpur, Gujarat state, India; (A) dorsal view; (B) ventral view male (CW: 29.98 mm); (C) ventral view ovigerous female (CW: 27.79 mm), (D) ventral view female (CW: 28.63 mm).** Scale bar 5 mm.

## Field methods

Monthly field work was conducted for 12 consecutive months from March 2021 to February 2022. The month wise data was compiled into winter season (November to February months), summer season (March to June months), and monsoon seasons (July to October months), following _Rao & Rama-Sharma (1990)_, to observe the seasonal variation. Catch-per-unit effort using the hand-picking method was used for the collection of specimens for a time period of 4 h at the time of low tide. When the water receded, a 500 m$^2$ area in the intertidal region was marked off and thoroughly examined for the presence of _L. exaratus_. Small rocks were also upturned for the presence of _L. exaratus_, which they prefer to occupy. Whenever an individual crab was encountered, the crab was collected and preserved in 10% formalin pending additional examination.

## Laboratory analysis

The crabs were identified on the basis of their morphological characters as follows using standard identification key provided by _Lee et al. (2013)_: Carapace is transversely sub ovate, and lightly granular. There are four large, triangular teeth on the anterolateral border, behind the exorbital angle. Male abdomen tapered, somites 3–5 fused, somite 6 elongated, 1.6 times longer than the telson. Chelipeds are unequal in size; fingers are stout with dark pigmentation excluding the tips, which are white in colour. Further, the individuals were categorised as male, non-ovigerous female, or ovigerous female (Fig. 2). For morphological character, carapace width (CW) was measured by digital vernier

callipers (Mitutoyo 500-197-20) (0.01 mm accuracy) and wet weight of crabs was measured using weighing balance (Sartorius–BSA224S–CW) (0.001 g accuracy).

The following method for fecundity study was adopted from *Patel, Vachhrajani & Trivedi (2023)*. Fecundity estimation conducted by cautiously taking out the mass of eggs present on the pleopods of ovigerous females ($n = 34$) and measuring three parameters *i.e.*, total number of eggs, weight of egg mass and size of eggs (diameter). For the total number of eggs, the egg mass was transferred into 20 ml of sea water and mixed gently so that the eggs got distributed evenly in the water. From this solution, three samples of 2 ml each were taken in a petri dish and observed under a stereo zoom microscope (Matlab–PST–901; Matlab, Natick, MA, USA) to count the total number of eggs. The total number of eggs in each sample was divided by three and multiplied by the dilution factor (10) to obtain the total number of eggs (*Litulo, 2004*). Ovigerous females were weighed both with and without egg mass, and the difference in their weight was considered as the weight of egg mass. Eggs ($n = 10$) from each ovigerous female were measured by means of an ocular micrometre under a microscope for the size range (*Saher & Qureshi, 2010*).

## Data analysis

### Population structure

The specimens were grouped in 2 mm size class intervals from 4 to 30 mm CW in order to get the overall size frequency distribution. Shapiro Wilk test was conducted to analyse the normality of the collected data, which suggests that the data distribution was not normal ($p < 0.001$). Hence, non-parametric analysis was carried out. To investigate the difference in the variance of mean values of the carapace width of male, non-ovigerous, and ovigerous individuals, a Kruskal-Wallis (KW) test was conducted. On getting a significant difference ($p < 0.05$) in the CW between the sexes, a multiple comparison analysis using Dunn's *post hoc* test was used to do a multiple comparison study. Monthly variations in the size (CW) and sex composition of *L. exaratus* individuals were obtained by plotting the data on individuals' carapace width and sex. The ratio of males and females (ovigerous and non-ovigerous females) was evaluated by the means of chi-square test ($\chi^2$). The size at first maturity was determined by calculating the percentage of ovigerous females across various size classes from the total number of samples collected. Juveniles were defined as individuals that were smaller than the smallest ovigerous female (*Baeza et al., 2013*). The effect of temperature on *L. exaratus* breeding and juvenile settling was examined by plotting monthly data on the incidence of juvenile and ovigerous females against ambient temperature. The relationship between the mean ambient temperature and the relative juvenile frequency was examined using Pearson's correlation analysis.

### Fecundity

To investigate the relationship between the morphological features of eggs (total number of eggs, egg mass weight and size of eggs) and crabs' morphology (CW and weight) regression analysis was performed. At $p < 0.05$, the statistical significance was deemed significant. Microsoft Excel and PAST software, version 4.03 (*Hammer, Harper & Ryan, 2001*), were used to carry out statistical analyses.

## RESULTS

During the study period, 1,215 individuals were investigated in total; 558 of them were male (45.93%) and 657 of them were female (54.07%) (Table 1). The carapace width of *L. exaratus* males ranged from 5.15 to 29.98 mm, while in case of females it ranged from 5.26 to 28.63 mm. The size differences between the male and female individuals were not statistically significant (Kruskal-Wallis, H = 0.209, $p$ = 0.646) (Table 1).

Table 2 shows that the year-round average total sex ratio (1:1.2) for *L. exaratus* was significantly different from the predicted 1:1 proportion ($\chi^2$ = 4.1219, $p$ = 0.042) and biased towards females. Month wise, female biased sex ratio was observed in almost all the months except September (1:0.8), October (1:0.9), and November (1:0.7). November had the highest percentage of male occurrences (57.89%), while April had the lowest rate (37.41%). In terms of females, the highest percentage of non-ovigerous female occurrences were observed in June (50%) and August (50%), while the lowest percentages were observed in January (27.18%). Ovigerous females were collected all year, which shows the species is breeding all year-round. However, from December to April, the greatest percentage of occurrence was recorded of ovigerous females, suggesting a peak in the breeding season.

The individuals of *L. exaratus* occurred in all the size classes between 4 to 30 mm. It was observed that males exhibited bimodal pattern of distribution having maximum occurrence in 6–8 mm CW size class and 24–26 mm CW size class. On the other hand, females exhibited unimodal pattern of size frequency distribution, with maximum occurrence recorded in 14–16 mm CW size class (Fig. 3).

Moreover, there was a considerable variation in the occurrence of adults, ovigerous females and juveniles (<12 mm) during different months of the year (Fig. 4). It was found that in April, May, June, and July (summer and early monsoon season) the population of juveniles was least as compared to the adult population. Moderately, less number of juveniles were also observed during December to March (winter and early summer season) than August to November (Monsoon and early winter season) as compared to adult male and female (Fig. 4).

Males had a bimodal distribution during most of the months, while non-ovigerous females showed a unimodal distribution pattern, as was also observed in ovigerous females. Furthermore, it was also observed that juveniles were present all year round (Figs. 5–7). A negative correlation (Pearson's correlation, r = −0.39) was observed between the mean ambient temperature and relative frequency of juveniles (Fig. 8).

The results of fecundity revealed that the CW of ovigerous females was between 10.38 and 24.02 mm, with their average size being 17.95 ± 3.81 mm ($n$ = 34). The wet body weight of the ovigerous females was recorded between 0.41 and 4.64 g, with the mean weight being 2.01 ± 1.1 g ($n$ = 34). The average number of eggs observed was 4,529 ± 2,003 ($n$ = 34), with the minimum and maximum reported being 920 and 8,730 eggs, respectively. The average egg size ($n$ = 34) was 0.36 ± 0.07 mm, with the minimum and maximum observed sizes being 0.19 and 0.54 mm, respectively. The average egg mass weight ($n$ = 34) was 0.29 ± 0.18 g, with the minimum and maximum observed egg mass

**Table 1 Carapace width values of male and female individuals of *L. exaratus* from Shivrajpur, Gujarat state, India.**

| Sex | *n* | Min. CW (mm) | Max. CW (mm) | Mean ± SD |
|---|---|---|---|---|
| Male | 558 | 5.15 | 29.98 | 15.967 ± 5.27* |
| Female | 657 | 5.26 | 28.63 | 15.48 ± 3.77* |

Notes:
*n*, total number of individuals; CW, carapace width.
* Significant level if $p < 0.05$ (*).

**Table 2 Total number of *L. exaratus* specimens collected from Shivrajpur, Gujarat state, India.**

| Month | M | % | NOF | % | OF | % | NOF+ OF | % | Sex ratio |
|---|---|---|---|---|---|---|---|---|---|
| January | 51 | 49.51 | 28 | 27.18 | 24 | 23.30 | 52 | 50.49 | 1:1.02 |
| February | 56 | 43.41 | 40 | 31.01 | 33 | 25.58 | 73 | 56.59 | 1:1.3 |
| March | 55 | 44.00 | 42 | 33.60 | 28 | 22.40 | 70 | 56.00 | 1:1.3 |
| April | 55 | 37.41 | 51 | 34.69 | 41 | 27.89 | 92 | 62.59 | 1:1.7 |
| May | 69 | 40.83 | 69 | 40.83 | 31 | 18.34 | 100 | 59.17 | 1:1.4 |
| June | 20 | 43.48 | 23 | 50.00 | 3 | 6.52 | 26 | 56.52 | 1:1.3 |
| July | 17 | 45.95 | 14 | 37.84 | 6 | 16.22 | 20 | 54.05 | 1:1.2 |
| August | 44 | 44.00 | 50 | 50.00 | 6 | 6.00 | 56 | 56.00 | 1:1.3 |
| September | 54 | 56.25 | 37 | 38.54 | 5 | 5.21 | 42 | 43.75 | 1:0.8 |
| October | 45 | 53.57 | 34 | 40.48 | 5 | 5.95 | 39 | 46.43 | 1:0.9 |
| November | 44 | 57.89 | 22 | 28.95 | 10 | 13.16 | 32 | 42.11 | 1:0.7 |
| December | 48 | 46.60 | 32 | 31.07 | 23 | 22.33 | 55 | 53.40 | 1:1.1 |
| Total | 558 | 45.93 | 442 | 36.38 | 215 | 17.70 | 657 | 54.07 | 1:1.2 |

Note:
M, Male; NOF, Non-ovigerous female; OF, Ovigerous female. Chi-square test ($\chi^2$) = 4.1219, $p$ = 0.042.

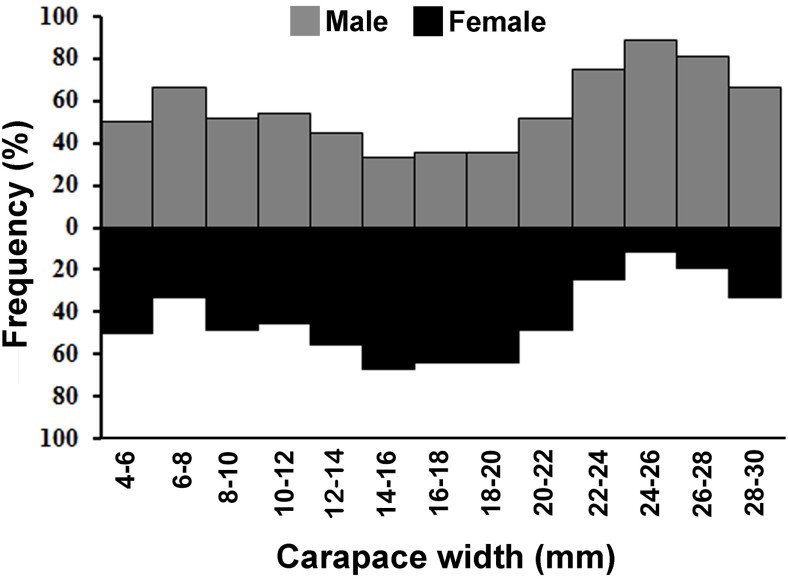

**Figure 3 Overall size frequency distribution of *L. exaratus* collected from Shivrajpur, Gujarat state, India.**

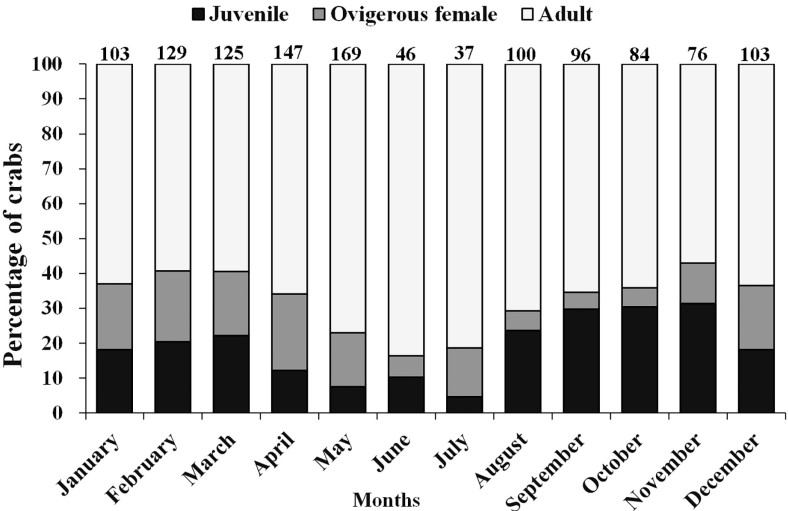

**Figure 4 Percentage of different demographic categories of *L. exaratus* from Shivrajpur, Gujarat state, India during the 12 months of study period.**

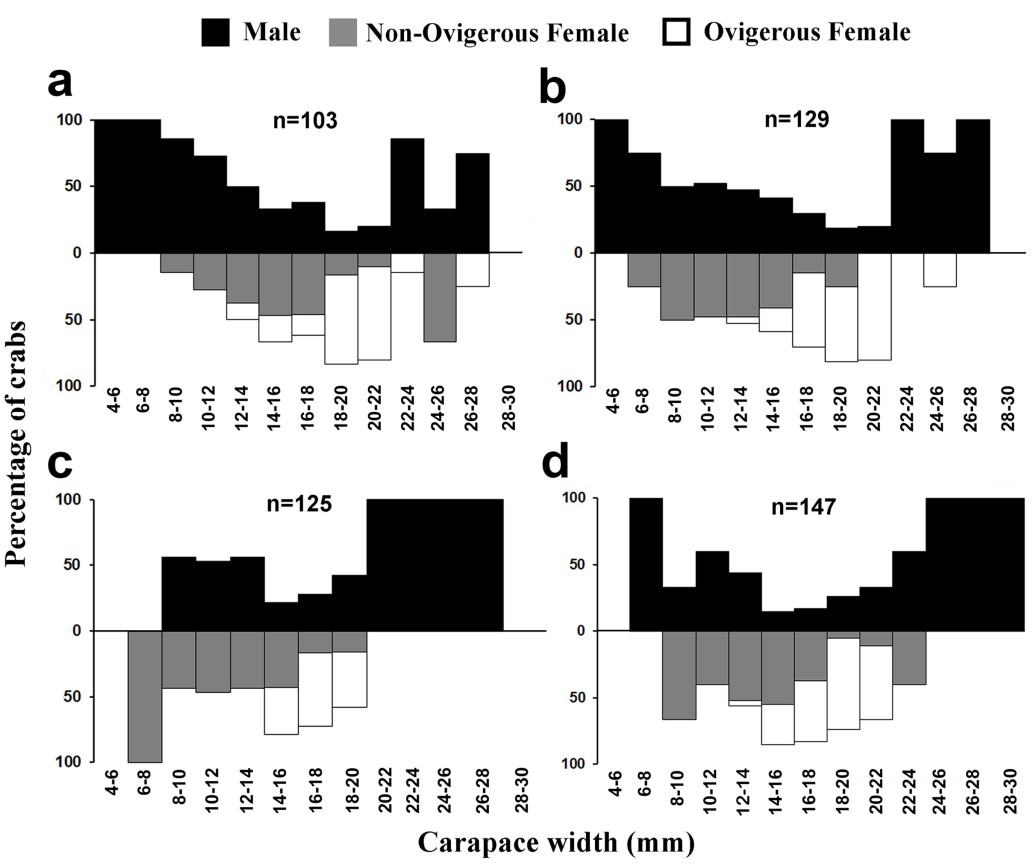

**Figure 5 Size frequency distribution of *L. exaratus* in each month from Shivrajpur, Gujarat state, India; (A) January, (B) February, (C) March, (D) April.**

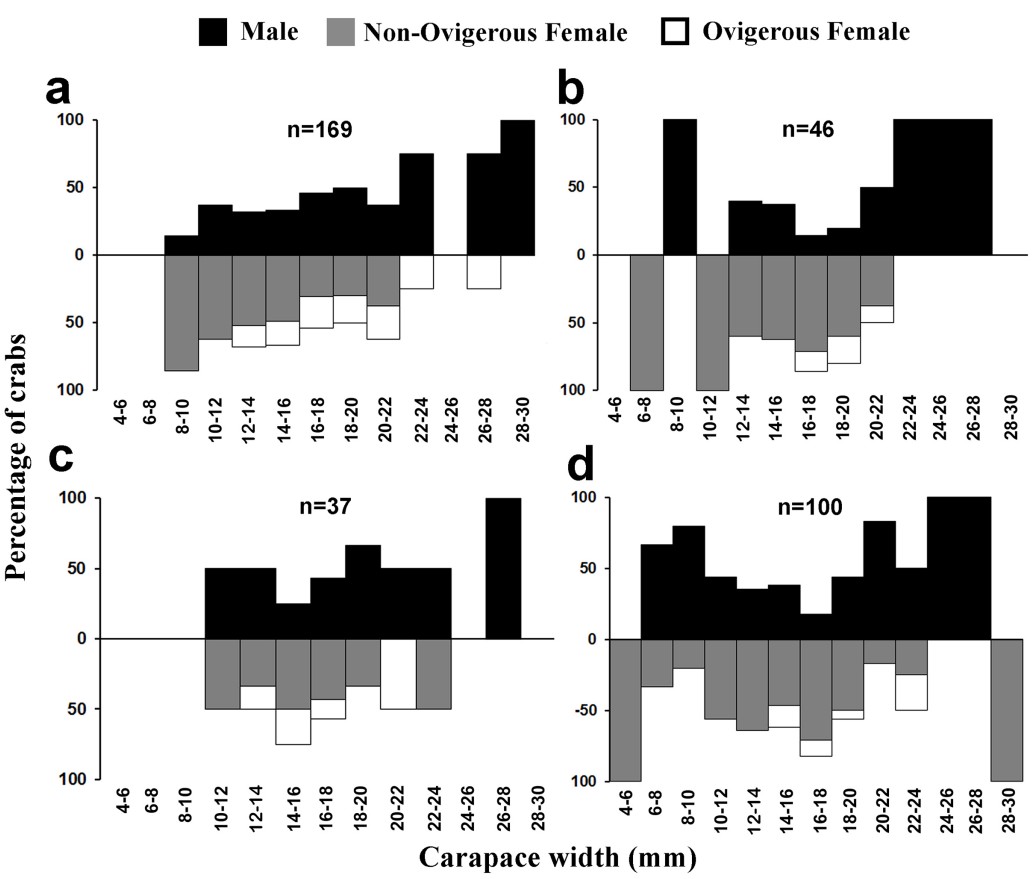

**Figure 6** Size frequency distribution of *L. exaratus* in each month from Shivrajpur, Gujarat state, India; (A) May, (B) June, (C) July, (D) August.

weights being 0.04 and 0.88 g, respectively (Table 3). The ovigerous females' carapace width and body weight were shown to be significantly correlated with both the egg weight and total number of eggs (Fig. 9).

## DISCUSSION

A significant variation was observed in the average carapace width of different sexes of *Dotilla blanfordi*, where it was found that male individuals were significantly larger than females. Studies conducted on the population structure of *Matuta planipes* and *Ashtoret lunaris* (*Saher et al., 2017*), *Uca bengali* (*Tina et al., 2015*), *Scylla olivacea*, *S. tranquebarica*, and *S. paramamosain* (*Waiho et al., 2021*), *Clibanarius ransoni* (*Patel, Patel & Trivedi, 2020*; *Patel, Vachhrajani & Trivedi, 2022*), *C. rhabdodactylus* (*Patel, Vachhrajani & Trivedi, 2023*), and *Diogenes custos* (*Patel et al., 2020*) also exhibited similar results. It has been observed that the growth rate of female individuals is generally reduced as a result of greater energy investment in gonadal development, which leads to decreased somatic growth in comparison to male individuals (*Mantelatto et al., 2010*). Another hypothesis suggests that the chances of attracting and obtaining females for the purpose of mating increases with increased size of male individuals (*Wada, Kitaoka & Goshima, 2000*), while

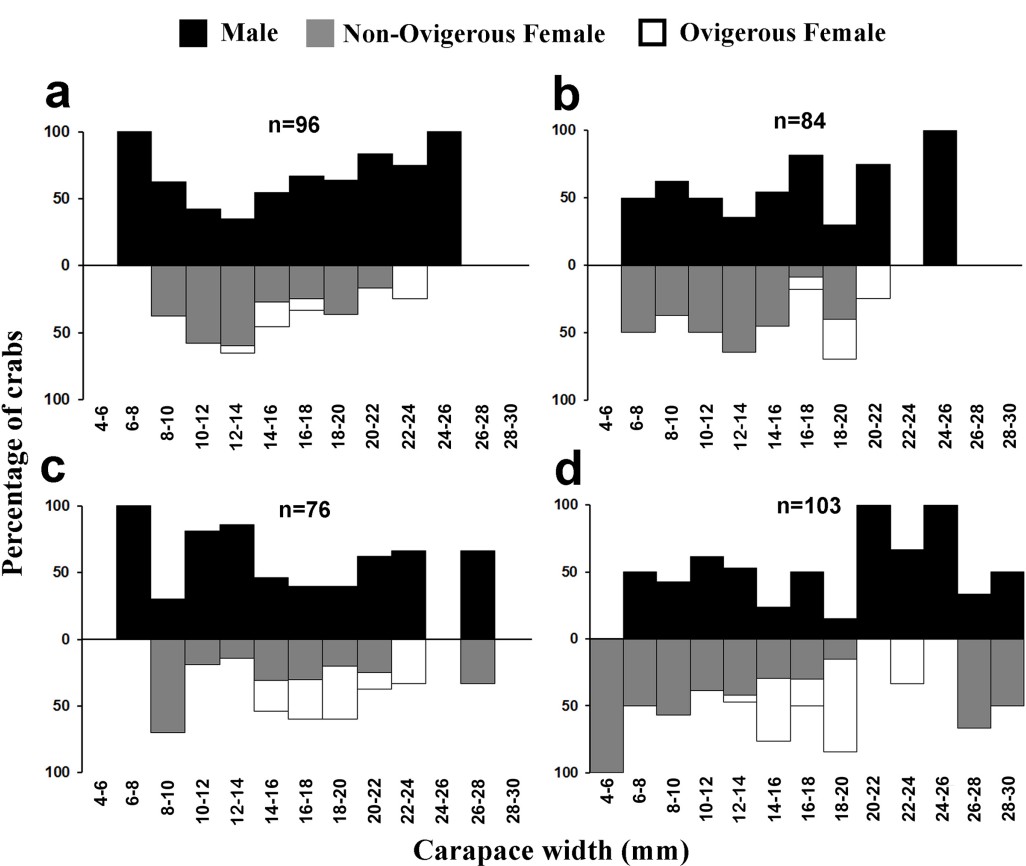

**Figure 7 Size frequency distribution of *L. exaratus* in each month from Shivrajpur, Gujarat state, India; (A) September, (B) October, (C) November, (D) December.**

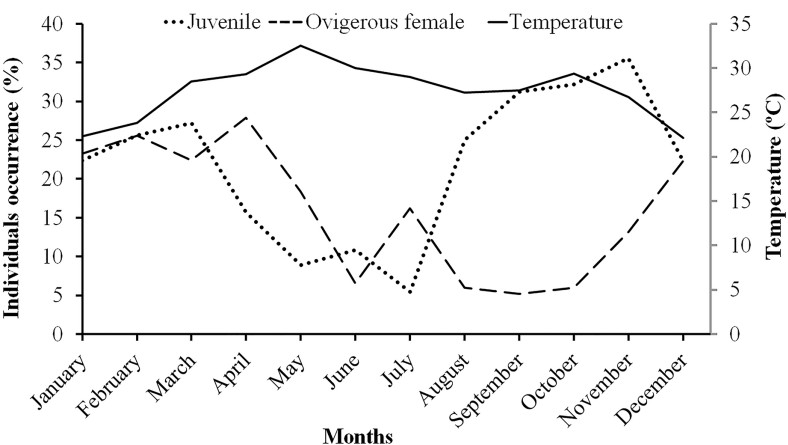

**Figure 8 Association between the juveniles (of both sexes) and ovigerous female occurrence of *L. exaratus* with monthly ambient temperature from Shivrajpur, Gujarat state, India.**

**Table 3 Summary of different morphological parameters of *L. exaratus* ovigerous females and eggs from Shivrajpur, Gujarat state, India.**

| Variables | *n* | Mean ± SD | Min. | Max. |
|---|---|---|---|---|
| Crab weight (g) | 34 | 2.04 ± 1.2 | 0.41 | 4.64 |
| Weight of egg mass (g) | 34 | 0.29 ± 0.18 | 0.04 | 0.88 |
| Carapace length (mm) | 34 | 12.14 ± 2.4 | 6.92 | 15.88 |
| Carapace width (mm) | 34 | 18.1 ± 3.8 | 10.38 | 24.02 |
| Egg number | 34 | 4,529 ± 2,003 | 920 | 8,730 |
| Egg size (mm) | 34 | 0.36 ± 0.07 | 0.19 | 0.54 |

**Note:**

*n*, total individuals; SD, standard deviation.

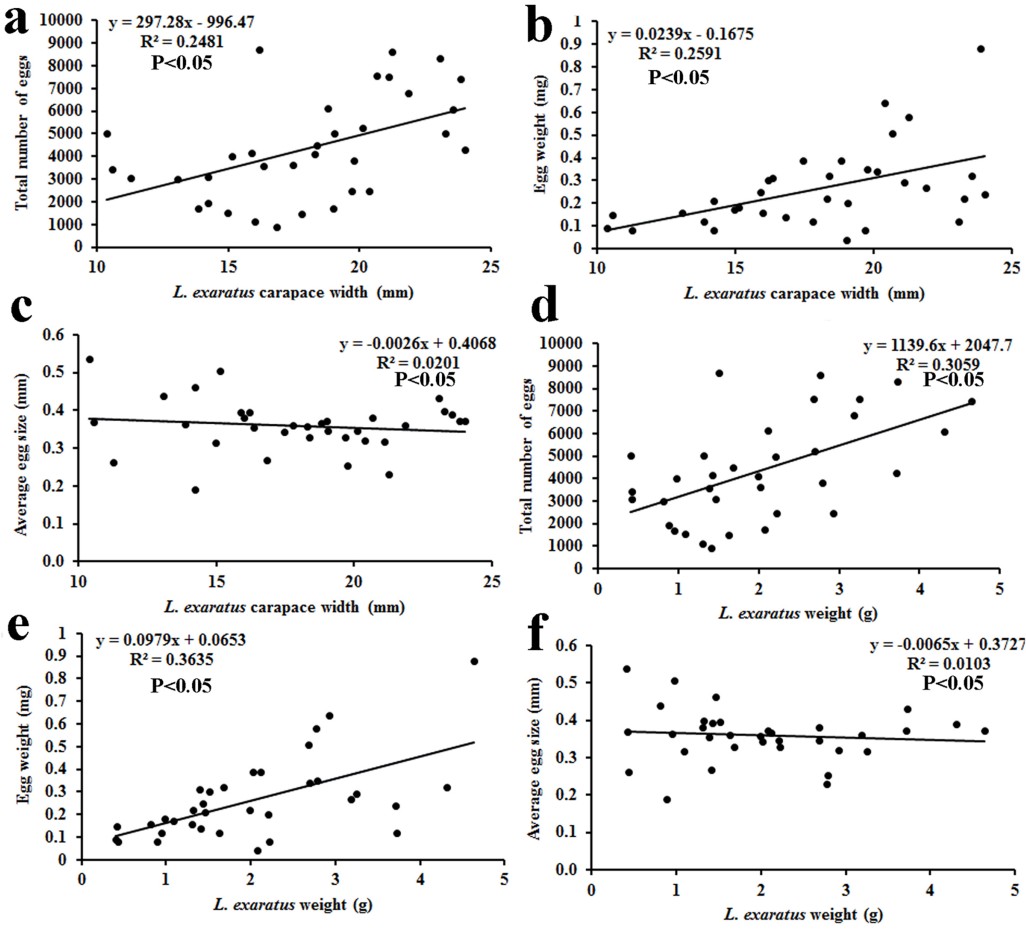

**Figure 9 Relationship of *L. exaratus* carapace width (mm) with (A) total number of eggs; (B) egg weight; and (C) average egg size; and crab weight (g) with (D) total number of eggs; (E) egg weight; and (F) average egg size.**

the difference in size also reduces intraspecific competition among different sexes for available resources (*Abrams, 1988*).

Overall sex ratio (1:1.2) was found to be female-biased, while month-wise also female biased sex ratio was observed except September, October and November months.

In general, natural selection promotes a sex ratio of 1:1 parental expenditure on offspring (*Taylor, 1996*); however, deviation from the ideal sex ratios is common in marine crustaceans, as observed in *Calcinus tibicen* (*Fransozo & Mantelatto, 1998*), *Limulus polyphemus* (*Smith et al., 2002*), *Crangon crangon* (*Siegel, Damm & Neudecker, 2008*), *Opusia indica* (*Saher & Qureshi, 2011*), and *Macrophthalmus* (*Venitus*) *dentipes* (*Qureshi & Saher, 2012*). The sex ratio also differed during different growth stages, with an ideal sex ratio (1:1) in smaller size classes (1–3 mm CW), female biased in intermediate size classes (3–6 mm CW) and exclusively male biased in larger individuals (6–8 mm CW). Certain other studies have found similar results (*Gherardi & Nardone, 1997*; *Bezerra & Matthews-Cascon, 2007*; *Mishima & Henmi, 2008*; *Manzoor et al., 2016*). Numerous factors can be responsible for such deviation in the sex ratio including competition in local mate (*Hamilton, 1967*), differences in the efficiency of utilizing local resources that biases sex ratios (*Silk, 1983*), difference in the investment in male and female offspring (*Kobayashi et al., 2018*), and sexual selection (*Swanson et al., 2013*). Sexual dimorphism in size could be one of the reasons for the different sex ratio from the ideal 1:1 in different size classes. Higher male mortality in the intermediate-size classes often leads to a female biased sex ratio (*Asakura, 1995*). Moreover, males grow to bigger sizes quickly than females, leading to male biased sex ratio on the larger size classes (*Wenner, 1972*). Disparities in sexual mortality and dispersion may potentially contribute to unbalanced sex ratios in crab populations (*Johnson, 2003*).

Present investigation found that the size frequency distribution of *L. exaratus* males had a bimodal distribution, while the females had a unimodal distribution. Also, there was a considerable difference in the seasonal size frequency distribution. Similar results have been observed in *Paguristes tortugae* (*Mantelatto & Sousa, 2000*), *Chaceon affinis* (*López Abellán, Balguerías & Fernández-Vergaz, 2002*), *Pilumnus vespertilio* (*Litulo, 2005*), *Dilocarcinus pagei* (*Taddei et al., 2015*) *Aegla georginae* (*Copatti et al., 2016*) and *Clibanarius rhabdodactylus* (*Patel, Vachhrajani & Trivedi, 2023*). Over time, the population size and frequency of dispersion may be significantly changed by the rapid recruitment of larvae and reproductive rate (*Thurman, 1985*). Such distributions have been explained by a variety of theories, such as differential patterns of migration (*Flores & Negreiros-Fransozo, 1999*), growth rate (*Negreiros-Fransozo, Costa & Colpo, 2003*), and differential death rate (*Díaz & Conde, 1989*). It is commonly found in organisms that undergo several rounds of reproduction and generate a large number of clutches every season (*Zimmerman & Felder, 1991*). Unimodality is often seen in stable populations that have approximately equal numbers of new members and emigrants, consistent recruitment and mortality rates throughout the course of the life cycle, and steady demographics (*Thurman, 1985*; *Díaz & Conde, 1989*) whereas, bimodality could be an indication of the general tendencies in population increase.

The ambient temperature of the study site ranged from 22.1 °C to 32.5 °C, which is within the range of a tropical environment that may support continuous reproduction. Hence, there was year-round occurrence of ovigerous females suggesting that *L. exaratus* is a continuously breeding species that has maximum recorded frequency from December to April. Similarly, studies carried out on *L. exaratus* (*Al-Wazzan et al., 2020*), *Scylla olivacea*

(*Ali et al., 2020*), *Opusia indica* (*Saher & Qureshi, 2011*), *Emerita portoricensis* and *E. asiatica* (*Goodbody, 1965*), *Ilyoplax frater* (*Saher & Qureshi, 2010*), *Diogenes brevirostris* (*Litulo, 2004*) and *Petrochirus diogenes* (*Bertini & Fransozo, 2002*) did not find any association between the frequency occurrence of ovigerous females and ambient temperature. As *L. exaratus* is common inhabitant of upper intertidal region where higher temperature during summer season can greatly increase the desiccation risk, leading to migration of ovigerous females in deeper water (*Allen, 1966*; *Asakura, 1987*; *Al-Wazzan et al., 2020*) resulting in decreased abundance in the intertidal region. As a result, seasonal fluctuations in abundance reflect both migration and mortality, while summer abundance estimates may underestimate the size of the local population. However, it was found that juvenile percentage occurrence increased with a decline in ovigerous female percentage occurrence, whereas juvenile percentage occurrence declined when ovigerous female percentage occurrence increased. Such outcomes demonstrate that the species may recruit juveniles throughout the year as a result of rapid reproduction and a short incubation time. Similar outcomes have been reported in several other studies including *Deiratonotus japonicus* (*Oh & Lee, 2020*), *Scylla olivacea* (*Rouf et al., 2021*), *Clibanarius rhabdodactylus* (*Patel, Vachhrajani & Trivedi, 2023*), *Dardanus deformis* (*Litulo, 2005*), and *Menippe nodifrons* (*Fransozo, Bertini & Correa, 2000*). There are a number of variables, including the availability of food for adults (*Goodbody, 1965*), the ecology of larvae (*Reese, 1968*), the amount of time to attain sexual maturity, the timing of mating and gonadal development, as well as the length of the incubation period (*Sastry, Vernberg & Vernberg, 1983*), which can lead to periodicity in the reproductive season. A variety of abiotic and biotic variables, including water temperature (*Chou, Head & Backwell, 2019*), salinity (*Huang et al., 2022*), the nutritional quality of the females (*Matias et al., 2016*), variations in photoperiod (*Zhang et al., 2023*), the amount and availability of nutrition (*ViñaTrillos, Brante & Urzúa, 2023*), and the threat of predation (*Touchon, Gomez-Mestre & Warkentin, 2006*), may affect the reproductive maxima among populations.

It was found that the CW and wet body weight of *L. exaratus* were having positive correlation with total number of eggs and egg mass weight. Several other studies have also found similar results (*Patel, Vachhrajani & Trivedi, 2023*; *Crowley et al., 2019*; *Hamasaki, Ishii & Dan, 2021*; *Aviz et al., 2022*; *Mustaquim, Khatoon & Rashid, 2022*). Additionally, it has been demonstrated that ovigerous females with the same CW had variations in the number of eggs, egg mass weight, and egg size resulting from variations in the food supply, variation in egg production, and egg loss (*Hines, 1982*).

## CONCLUSIONS

The goal of the current study was to better understand the population structure and breeding biology of *L. exaratus*. Significant sexual dimorphism was found, with males being larger than females, most likely as a result of the size of gamete formation differing between the sexes and females investing more in egg production. Total sex ratio of species was 1:1.2 and monthly populations may be a result of differential biology and behaviour as well as the impact of biotic and abiotic variables on male and female individuals. The year-round occurrence of ovigerous females suggests continuous breeding of the

population and an inverse relationship between the peak in juvenile recruitment and the occurrence of ovigerous females which is a common phenomenon of tropical brachyuran crabs. There was a positive correlation between the egg parameters (weight of egg mass and number of eggs) and the morphology of ovigerous females (carapace width and body weight). Fecundity may be impacted by a variety of internal and external variables, such as the amount of energy used for somatic development and egg production. The present study was conducted at Shivrajpur village, a renowned tourist site with a blue-flag beach where various water sports activities take place. These activities along with higher tourist rush at the study site that may impact the habitat composition of the coast and also potentially influencing the ecology of *L. exaratus*. Furthermore, our findings will contribute to understanding the species' response to environmental changes, as both population structure and fecundity are closely tied to environmental variables.

## ACKNOWLEDGEMENTS

All the authors are thankful to Dhruva Trivedi for technical assistance in field work.

### Funding

The International Association for Biological Oceanography (IABO) funded the APC of this article. The funders had no role in study design, data collection and analysis, decision to publish, or preparation of the manuscript.

### Competing Interests

The authors declare that they have no competing interests.

### Author Contributions

- Krupal Patel conceived and designed the experiments, performed the experiments, analyzed the data, prepared figures and/or tables, authored or reviewed drafts of the article, and approved the final draft.
- Heris Patel performed the experiments, analyzed the data, prepared figures and/or tables, and approved the final draft.
- Swapnil Gosavi performed the experiments, authored or reviewed drafts of the article, and approved the final draft.
- Kauresh Vachhrajani conceived and designed the experiments, authored or reviewed drafts of the article, and approved the final draft.
- Jigneshkumar Trivedi conceived and designed the experiments, authored or reviewed drafts of the article, and approved the final draft.

### Data Availability

The raw data used for analysis are available in the Supplemental File.

## Supplemental Information

Supplemental information for this article can be found online at http://dx.doi.org/10.7717/peerj.16916#supplemental-information.

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
