# Peer review of "Population structure and fecundity of the Xanthid crab Leptodius exaratus (H. Milne Edwards, 1834) on the rocky shore of Gujarat state, India"

_PeerJ, doi:10.7717/peerj.16916_

## Round 0.1 · original submission · Major Revisions

Your manuscript has been reviewed and referees pointed out various details in need to attention. Please, I strongly suggest to carefully follow those details, such as the justification of the study, ecological aspects of the crab species, size frequencies precisely defined, describe the methodology better, figure appearance and please, English grammar needs improvement.

**Language Note:** The Academic Editor has identified that the English language must be improved. PeerJ can provide language editing services - please contact us at copyediting@peerj.com for pricing (be sure to provide your manuscript number and title). Alternatively, you should make your own arrangements to improve the language quality and provide details in your response letter. – PeerJ Staff

Reviewer 1 ·

Basic reporting

The author have been report the result follow from their objective, but they have to revised some issue that I wonder for fullfill this article to publish. such as some methodology. Moreover, the authtor also edit the reference in content to relate behind full reference.

Experimental design

I wonder about the fishing gear or something right that such handling and may be the author should be give the detail about the specimen after field collecting such as identified by morphology or other method such as dana barcoding. In addition to please give the detail about season and insert crab picture including ovigerous and non ovigerous in this article.

Validity of the findings

The author should be show the statistic analysis in some issue such as sex ratio and may be edit some value in the table because it's not related.

Additional comments

The author should be comply the content in part methodology related with their objective because now this section make the reader confuse.
Please give the detail for utilisation for this article to support diversity and environmental in the coastal area of India particulary in Gujarat state.

Annotated reviews are not available for download in order to protect the identity of reviewers who chose to remain anonymous.

Reviewer 2 ·

Basic reporting

This manuscript presents the population structure and fecundity of a Xanthid crab Leptodius exaratus in India.

This study is interesting and deserves to be published, but a few modifications are necessary beforehand.

1/ Figures are not uniform and should be sharper. They look like screenshots...

2/ English must be validated by an English speaker

Experimental design

The methods are consistent, but the following are missing

1/ GPS coordinates of the study site,
2/ Species names are not italicized (also throughout the document).
3/ add bibliographical references on the dip-net capture method, which seems to me to possibly create a bias in the capture of individuals

Validity of the findings

1/ In the introduction, we do not understand why it is interesting to study this species. I would suggest that the authors add more information on the ecological role of this crab and how climate change or human stress factors could have an effect on this species.

2/ The results are interesting, although I would suggest that size frequencies be more precisely defined. I don't quite agree with the graphs in figure 4, for example, because I suppose that size frequencies ranging every 5 mm (5-10; 10-15, etc.) are too large and we lose finer information on the different sizes. I'd suggest refining the scale every 1 mm or 2 mm, your work will be more interesting and powerful.


3/ and then to use more finesse - your discussion will be more powerful and striking, because your study is very interesting.

---

## Round 0.2 · Minor Revisions

Please, follow recommendations provides by the reviewer.

Reviewer 1 ·

Basic reporting

The author has been edited follow my suggestion but there are some point that they have to check such as 1) please consider the keyword should be related with the content 2) unit of egg mass changed from mm to g 3) please check the citation in the content and reference behind the paper should be relate and exactly. 4) scientific name of the carb should be italic style throughout the document 5) please edit the pattern the explaination of the figure in your paper 6) please check the format of the reference, there are many wrong reference such the author use the capital letter in the first in the title some reference etc.

Experimental design

The author was seperated the Experimental design followed my suggestion. The author reviwed the external morphology of the crab from the previous study but the author should explain the external morphology in brieftly and should be give the detail only the dominance characteristics. (Line 100-109)

Validity of the findings

please check the validity of technical term and edit the reference followed my suggestion in attached file.

Additional comments

please edit especially in part Reference and method section.

Annotated reviews are not available for download in order to protect the identity of reviewers who chose to remain anonymous.

---

## Round 0.3 · accepted · Accept

I have the following suggestions to improve your manuscript:

Title: Instead of ...a Xanthid Crab....it should the...the Xanthid Crab...
Abstract, bacground: The population structure and breeding biology of the Xanthid Crab on the rocky intertidal region of Shivrajpur in Saurashtra coast, Gujarat state, were examined.

Abstact, method: From March 2021 to February 2022, monthly sampling was conducted...